# Evaluation of a new implant for Tibial Tuberosity Transposition in dogs: An ex vivo study

Gabriel Rampanelli[1], Olicies da Cunha[1], Anderson Luiz de Carvalho[1], Cássio Ricardo Auada Ferrigno[2]*, Camila Aparecida Luiz[1‡], Laura Ayala Lazarotto[1‡], Lucas Dill Mocellin[1‡], João Pedro Cosmo Machado[1‡], Fernando Lunardelli[3‡]

1 Department of Animal Science, Federal University of Paraná, Paraná, Brazil, 2 Department of Small Animal Clinical Sciences, University of Tennessee, Knoxville, Tennessee, United States of America, 3 CDVET – Veterinary Diagnostic Center, Maringá, PR, Brazil

☙ These authors contributed equally to this work.
‡ CAL, LAL, LDM, JPCM and FL also contributed equally to this work.
* cferrign@utk.edu

## Abstract

The present study aims to describe and evaluate the effectiveness and potential applicability of a new implant designed to transpose the tibial tuberosity. The implant consists of a customized plate for fixation on the tibia, containing a hole for the insertion of a transposition screw in the tibial tuberosity. Computed tomography was performed on 21 cadaveric canine tibias to plan the surgical technique and calculate the desired transposition. Subsequently, an osteotomy of the tibial tuberosity was performed, extending 60% of the lateral cortex and 80% of the medial cortex to maintain a bone bridge between the tibial shaft and the osteotomized segment. After the osteotomy, the implant was fixed to the tibia, and the tibial tuberosity was transposed slowly and gradually using the transposition screw. The samples underwent radiographic evaluation and manual palpation following the application to detect tibial crest fractures. The plate's shape adequately fit the medial surface of the tibia, and the implant was effective in promoting a slow and gradual transposition in canine cadavers without the need for an additional surgical device to perform the maneuver. The implant proved to be effective in achieving the desired transposition in a progressive, gradual, and slow manner.

## Introduction

Medial patellar luxation is a common orthopedic condition in dogs, and it can be of either congenital or traumatic origin [1–3]. It is characterized by improper positioning of the patella medially to the trochlear groove [2,4,5].

Various bone deformities can alter the alignment of the quadriceps extensor mechanism, leading to the development of medial patellar luxation. Among these, deviation of the tibial tuberosity is frequently observed [6].

**Data availability statement:** All relevant data are within the manuscript and its Supporting information files.

**Funding:** Engevet partially funded the research, which included the manufacturing of the implant and the instruments used. This study was financed in part by the Coordenação de Aperfeiçoamento de Pessoal de Nível Superior - Brasil (CAPES) – Finance Code 001. Author GR received a research fellowship from CAPES, which supported the development of this study. The funders had no role in study design, data collection and analysis, decision to publish, or preparation of the manuscript.

**Competing interests:** I have read the journal's policy and the authors of this manuscript have the following competing interests: Rampanelli, Cunha and Carvalho are the creators of the implant, nonetheless, the authors take all the necessary steps to ensure that there is no bias in the manuscript.

Surgical treatment is recommended in dogs with lameness due to patellar luxation [2,3,7]. Several techniques exist for correcting patellar luxation and the quadriceps mechanism. Common procedures include soft tissue imbrication, tibial tuberosity transposition (TTT), trochleoplasty, corrective osteotomies of the femur and tibia, and trochlear prosthesis [2,8].

The TTT is an effective technique for realigning the quadriceps extensor mechanism and can be combined with other procedures [9,10]. However, surgical complications arising from the correction of patellar luxation using TTT are reported in approximately 13% to 48% of cases [8,11–15]. Among these, patellar reluxation stands out as a significant complication, occurring in up to 19.8% of dogs, in addition to other adverse events such as implant failure or migration and tibial tuberosity avulsion [11,16].

In the last decade, new implants have been developed for fixation after the TTT, such as pins or screws adjacent to tibial tuberosity [17,18], locking plates [15], plates with spacers [11] and hemicerclages [16], however none of these implants have the capacity to displace the tibial tuberosity and require an additional device or manual pressure to perform the transposition. Therefore, an implant applied after an osteotomy that allows transposition in a controlled manner can be desirable.

This study aims to describe the RTP (Rampanelli Transposition Plate) implant developed by the authors in collaboration with the company Engevet (Engevet Ltd, Paraná, Brazil) and to evaluate the TTT technique using the developed implant. We hypothesize that the implant can facilitate tibial tuberosity transposition in a slow, progressive, and gradual manner.

## Materials and methods

### Study model and inclusion criteria

The study was submitted to and approved by the Animal Use Ethics Committee of the Federal University of Paraná – Palotina Sector (protocol no. 22/2023). Twenty-one tibias from adult dogs, aged between 1 and 12 years and weighing between 4 and 25 kg of small to medium-sized breeds, regardless of sex, were selected. The animals died from causes unrelated to this study and were sourced from the Pathology Service of the Federal University of Paraná – Palotina Sector. Each sample was radiographed in craniocaudal and mediolateral projections, wrapped in gauze soaked in 0.9% sodium chloride solution, and placed in individually labeled plastic bags and frozen at −18 °C. The inclusion criteria were bones without signs of fractures, bone callus, neoplasms, or open growth plates. Bones with iatrogenic damage, such as osteotomy errors, were excluded. Before the procedure, the samples were thawed at 6ºC for 12 hours.

### Implant description

**Plate.** The RTP consists of a set of implants comprising a plate for fixation on the tibia and a specific screw to perform the TTT (Fig 1A and 1B).

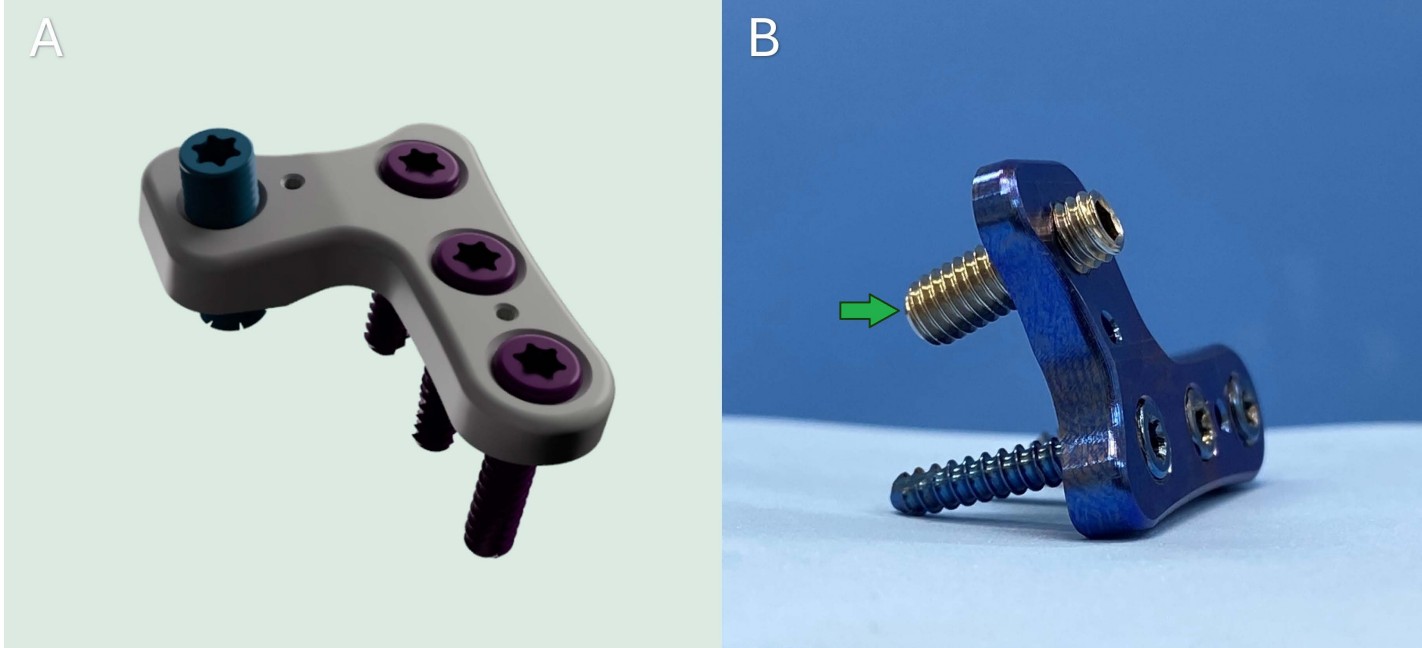

**Fig 1. RTP implant developed for the tibial tuberosity transposition technique.** (A) Three-dimensional visualization following the rendering of the implant. (B) Precision-machined implant for tibial tuberosity transposition. Green arrow: Transposition screw—a headless screw with a flat tip and full-length threading, designed to engage with the threaded hole of the plate. The rotation of this screw facilitates controlled displacement of the tibial tuberosity in accordance with the desired correction.

A custom inverted 'L'-shaped locking plate was created using Autodesk Fusion 360 (Autodesk Inc., San Francisco, USA). This plate features three threaded holes for tibial fixation with locking screws, two additional holes for temporary fixation with Kirschner wires, and one threaded hole designated for the transposition screw. The design allows for secure attachment of the implant to the tibial shaft, with one designated hole strategically positioned on the tibial tuberosity to improve the transposition process.

The transposition screw was developed as a headless and fully threaded screw with a flat tip to allow the gradual displacement of the osteotomized bone fragment, without interference from locking between the screw head and the plate surface, unlike conventional locking screws. These screws vary in length (4–10 mm), depending on the desired degree of transposition (Fig 1B). The transposition screw is available in two sizes to fit all plate sizes: M3x0.5 (3 mm diameter, 0.5 mm thread pitch) and M4x0.7 (4 mm diameter, 0.7 mm thread pitch).

Considering the wide range of sizes among dogs, five implant sizes were recommended, with distinct variations for the right and left sides. After the design process, both the plate and fixation screw were made from titanium (ASTM F163), while the transposition screw was made from 316L stainless steel.

**Osteotomy guide.** The osteotomy guide was engineered utilizing CAD Fusion 360 software (Autodesk Inc., San Francisco, USA) (Fig 2A and 2B). It incorporates both a proximal and a distal aperture designed for 0.8 mm Kirschner wires, facilitating temporary fixation to the tibia. Additionally, it features a 1.2 mm high slot designated for blade insertion at the osteotomy site, effectively preventing potential displacement during the procedure. Four distinct sizes were developed, with varying slot lengths of 15, 25, 35, and 45 mm.

The guide was produced using an Ender 3 3D printer (Creality 3D, Shenzhen, China) with polylactic acid (PLA) filament, in 0.2 mm layers and 100% infill.

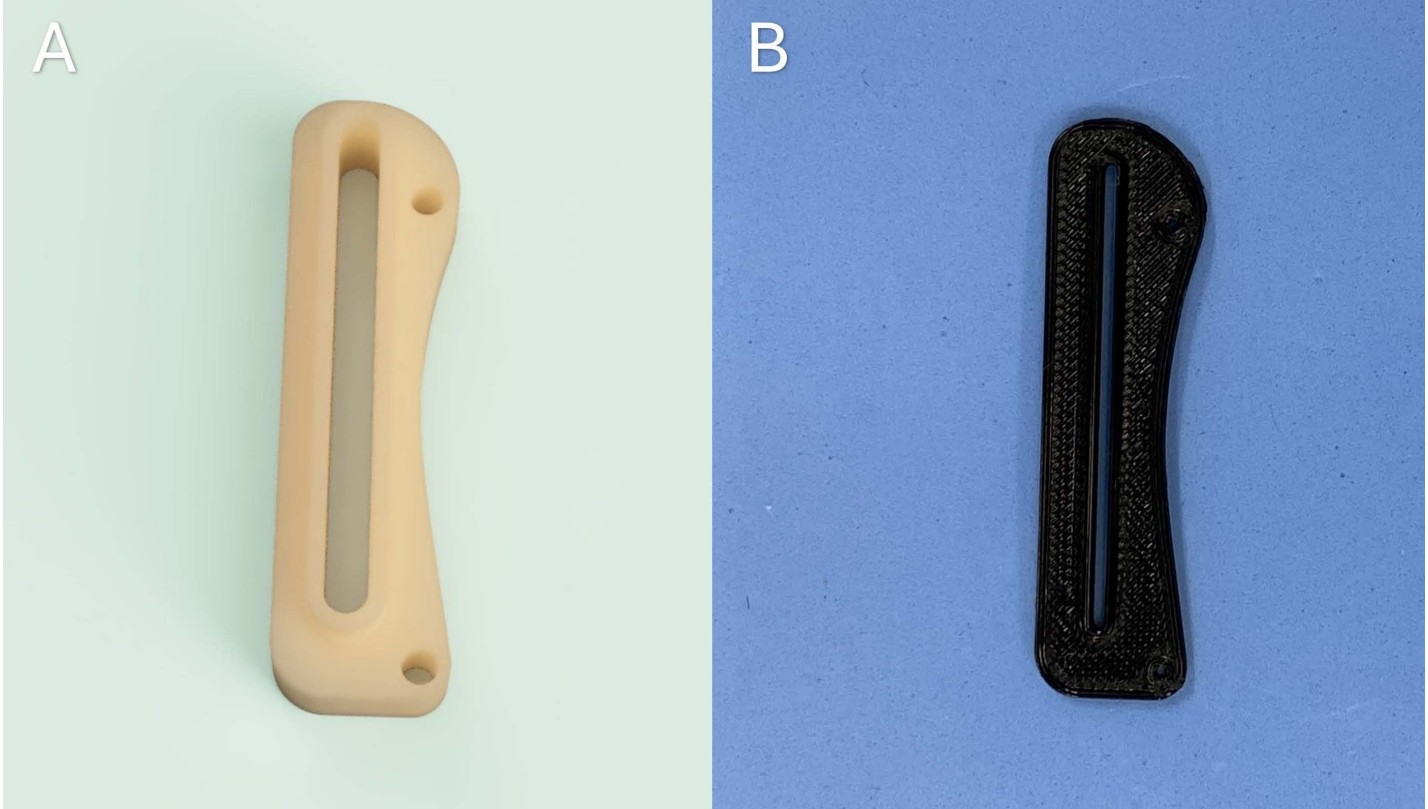

**Fig 2. Osteotomy guide developed for executing the osteotomy during the tibial tuberosity transposition procedure.** (A) Three-dimensional visualization following guide rendering. (B) Osteotomy guide subsequent to production.

## Surgical planning

All tibias underwent computed tomography (BrightsSpeed, 16 channels, GE HealthCare, Illinois, USA). The images were acquired in 0.6 mm slices and three-dimensionally reconstructed using a bone window (WL 287; WW 332) to perform the preoperative planning for TTT of each sample.

To calculate the displacement of the tibial tuberosity using the transposition screw, the transposition distance was standardized at one-half of the width of the tibial tuberosity, measured at the level of the patellar tendon insertion in the frontal plane.

The tibial mechanical axis was initially delineated in the medial view of the surgical plan within the vPOP Pro (VETSOS Education Ltd. – Shrewsbury, United Kingdom) software. Subsequently, a perpendicular line to the mechanical axis was established at the insertion of the patellar tendon, and an additional line was drawn from the distal limit of the tibial crest to the caudal cortex, which was also perpendicular to the mechanical axis. To accurately identify the tibial crest and the osteotomy region, a red line was drawn, aligned with Gerdy's tubercle and extending distally to bisect the tibia into 30% cranial and 70% caudal proportions, employing the white lines as reference points (Fig 3).

The tibial crest measurement determined values corresponding to 80% of the osteotomy line on the medial surface and 60% on the lateral surface. These represent the limits of the osteotomy on the medial and lateral cortices, respectively. This proportion enables transposition without requiring a complete tuberosity osteotomy [17].

Implant size selection was based on its dimensions, obtained from technical drawings, and on measurements of the medial tibial aspect, made from tomographic images. These measurements were performed using vPOP Pro (VETSOS Education Ltd. – Shrewbury, United Kingdom) software.

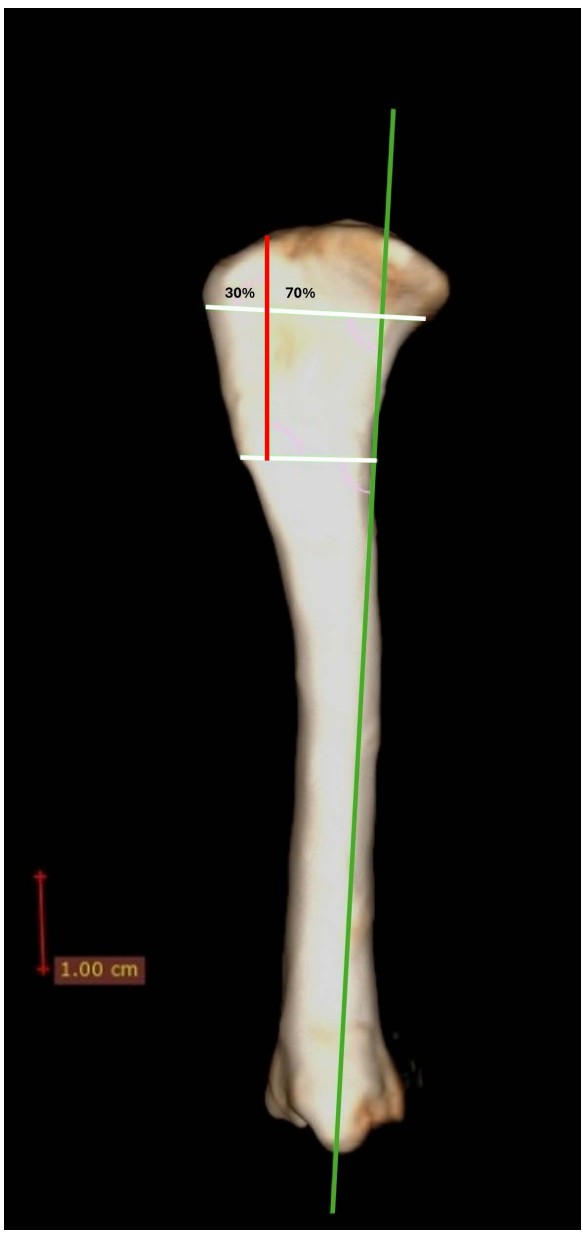

**Fig 3. Planning of the Tibial Tuberosity Transposition Procedure in vPOP Pro (VETSOS Education Ltd. – Shrewbury, United Kingdom).** Green line – tibial mechanical axis. White lines – orientation lines for locating the osteotomy site. Red line – tibial tuberosity osteotomy line.

## Implant application

The osteotomy guide was positioned on the medial aspect of the tibia using two Kirschner wires so that the osteotomy line aligned with Gerdy's tubercle, making the osteotomized tibial crest 30% of the tibial width in the sagittal plane (Fig 4A). Subsequently, an incomplete osteotomy was performed. The medial cortical osteotomy extended 80% of the tibial crest, while the lateral cortical osteotomy reached 60% of the length of the tibial crest, keeping the distal portion of the tibial crest connected to the tibial diaphysis. The osteotomy length for both cortices was measured using a Castroviejo caliper.

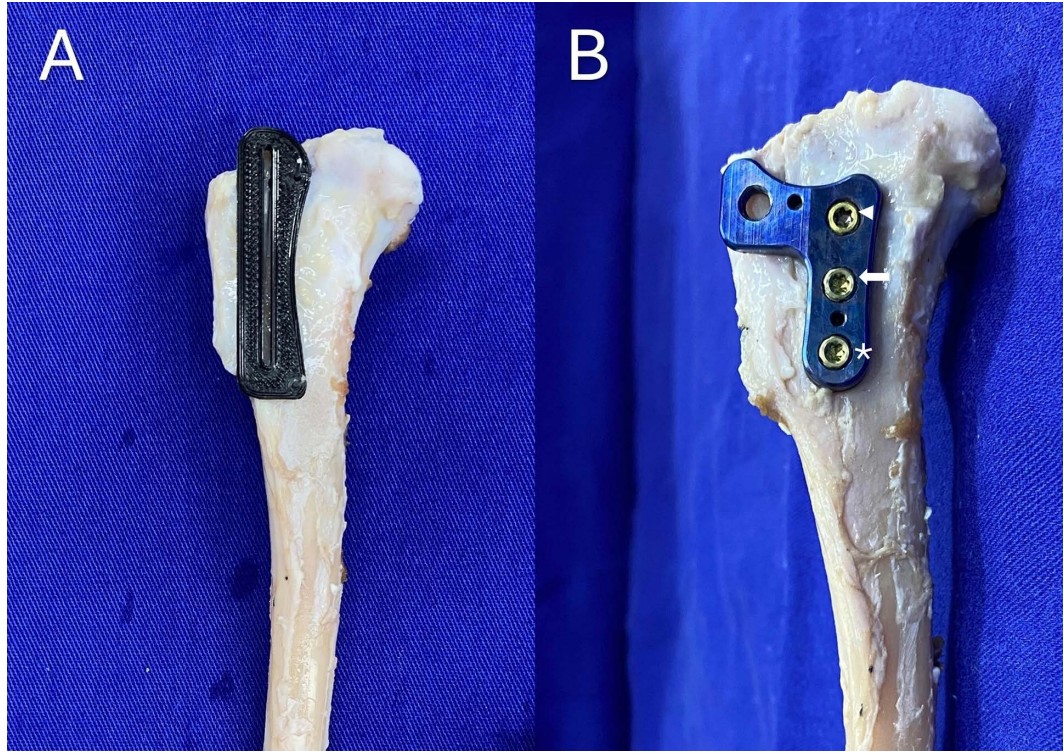

**Fig 4. (A) Positioning of the osteotomy guide with two Kirschner wires.** (B) Fixation of the RTP implant at the height of the patellar tendon insertion with three locking screws, inserted in sequential order: proximal (arrowhead), distal (asterisk), and central (arrow).

Following the osteotomy, the implant was secured to the medial aspect of the tibia, ensuring that the insertion site for the transposition screw was aligned with the patellar tendon insertion. To achieve this, the proximal locking screw was initially inserted, succeeded by the distal locking screw, and lastly, the central locking screw (Fig 4B). Subsequently, the transposition screw was inserted into the designated threaded hole and advanced until the tibial tuberosity reached the pre-measured displacement as delineated in the surgical plan (Fig 5A–5D). This process was executed slowly and gradually, at a rate of 0.7 mm/min, accompanied by a rotational speed of 1.4 revolutions per minute (RPM) for the M3x0.5 transposition screw and 1 RPM for the M4x0.7 screw (S1 Video) [17,18]. The displacement was verified using a caliper, and the transposition time was recorded with a stopwatch.

## Evaluation

Upon completion of the implant application, the samples underwent visual and tactile inspection and radiographic evaluation (Diafix Ltd, 500mA/125 kV - São Paulo, Brazil) in both mediolateral and craniocaudal projections. This was conducted to verify the displacement of the tibial tuberosity, identify any potential fractures of the tibial tuberosity, and assess the positioning of the implant and screws.

## Results

Of the 21 tibias, 17 had a 1.5 mm implant inserted, while 4 had a 2.7 mm implant, depending on bone size. All tibias had sufficient bone stock for implant fixation using three locking screws, along with space for the transposition screw to facilitate the tuberosity transposition near the patellar tendon insertion.

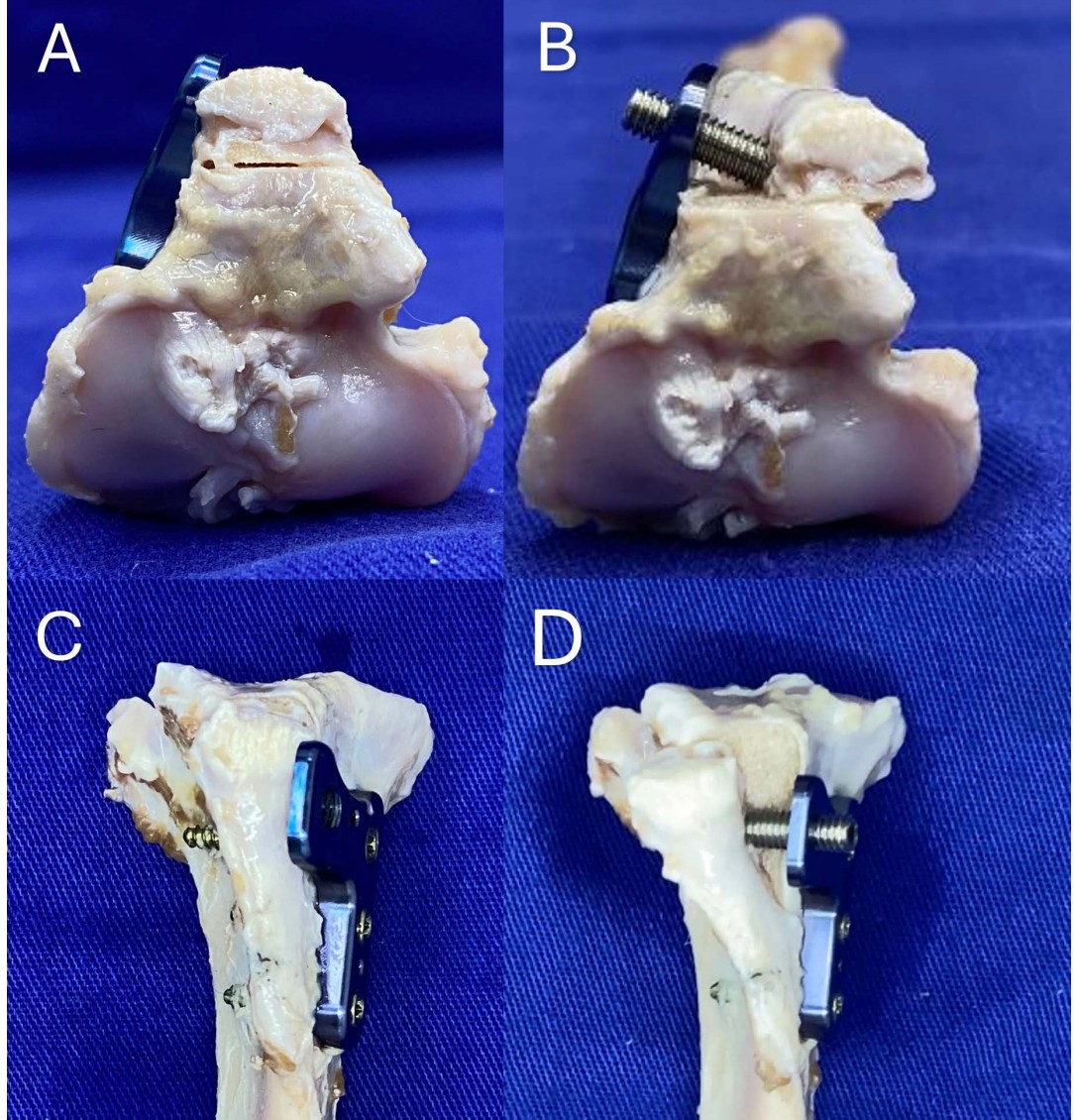

**Fig 5. (A) Proximal view of the tibia with the RTP implant, before performing the tibial tuberosity transposition.** (B) Proximal view of the tibia with the RTP implant after transposition. (C) Cranial view of the tibia with the RTP implant, before performing the tibial tuberosity transposition. (D) Cranial view of the tibia with the RTP implant after transposition.

Transposition required to achieve an equivalent to one-half of the width of the tibial tuberosity varied from 3.4 to 8.0 mm (mean: 5.7 ± 1.32 mm), and the transposition time varied from 4 min 55 s to 11 min 31 s (mean: 8 min 10 s ± 112 s) (Table 1). All tibias from cadavers reached the stipulated transposition in a slow, gradual, and controlled manner. Furthermore, appropriate positioning of the transposition screw was observed near the insertion of the patellial ligament. All specimens underwent visual and tactile inspection after the intended displacement, and no mobility suggestive of fracture was detected.

Radiographic evaluation in craniocaudal and mediolateral projections of the operated tibias showed no fissures or fractures associated with the TTT (Fig 6). Additionally, proper positioning of the screws and plate was observed, as well as the displacement of the tibial tuberosity.

**Table 1. Measurements of distance and transposition time associated with use of the RTP Implant for Tibial Tuberosity Transposition.**

| Tibia | TTW[a] | Transposition (mm) | Transposition Time | TTW[a]/ transposition |
|---|---|---|---|---|
| 1 | 10.1 | 5.9 | 8min27s | 0.59 |
| 2 | 10.3 | 6.0 | 8min36s | 0.58 |
| 3 | 9.9 | 5.8 | 8min25s | 0.60 |
| 4 | 10.8 | 5.5 | 7min58s | 0.52 |
| 5 | 6.2 | 3.6 | 5min14s | 0.59 |
| 6 | 6.4 | 3.4 | 4min55s | 0.54 |
| 7 | 8.1 | 4.2 | 6min06s | 0.53 |
| 8 | 9.3 | 4.3 | 6min16s | 0.47 |
| 9 | 15.7 | 8.0 | 11min28s | 0.51 |
| 10 | 15.5 | 8.0 | 11min31s | 0.52 |
| 11 | 8.8 | 5.1 | 7min21s | 0.58 |
| 12 | 6.4 | 3.7 | 5min18s | 0.58 |
| 13 | 11.4 | 6.3 | 9min07s | 0.56 |
| 14 | 11.2 | 6.3 | 9min01s | 0.56 |
| 15 | 13.1 | 7.3 | 10min32s | 0.56 |
| 16 | 13.1 | 7.3 | 10min28s | 0.56 |
| 17 | 9.7 | 5.0 | 7min13s | 0.52 |
| 18 | 9.9 | 5.2 | 7min27s | 0.53 |
| 19 | 12 | 6.3 | 9min01s | 0.53 |
| 20 | 11.9 | 6.2 | 8min57s | 0.53 |
| 21 | 10 | 5.7 | 8min17s | 0.58 |
| **Mean + SD** | **10.5 ± 2.6** | **5.7 ± 1.32** | **8min10s ± 112s** | **0.55 ± 0.03** |

[a]Tibial Tuberosity Width.

## Discussion

This study presents findings on the application of the newly developed RTP implant for TTT in tibias sourced from canine cadavers. The outcomes of this ex vivo investigation substantiated the effectiveness of the RTP implant in executing the transposition maneuver in a slow, progressive, and gradual manner, thereby corroborating the hypothesis. The technique employed an incomplete osteotomy, aimed at preserving a bony bridge in the distal region [17], while the implant maintains the lateral deviation of the newly transposed bone fragment. Additionally, the broader bone area affords enhanced resistance to the bending forces exerted during tibial tuberosity displacement, which may necessitate a greater applied force to transpose the osteotomized segment. Considering this information, the transposition screw proved effective in facilitating the displacement of the tibial crest.

The developed implant allows for precise, slow, and gradual transposition of the tibial tuberosity, as recommended by Sullivan et al, who established the value of 0.7 mm/min [17]. The deformation behavior under tension depends on the speed at which the force is applied; therefore, it is recommended that the transposition be performed slowly and continuously to prevent the occurrence of fractures, respecting the yield point and the elastic properties of the bone [17,18]. During the TT transposition phase, it is possible to perform stifle extension and flexion to ensure the correct alignment of the patella.

Literature reports describe the stabilization of TTT using plates, notably involving fixation of the plate into the tibial crest [11,19]. One example is the Rapid Luxation System (RLS) implant (Rita Leibinger Medical, Tuttlingen, Germany) [11], which employs a spacer secured with screws to the tibial crest to maintain the transposition. In contrast, the RTP implant

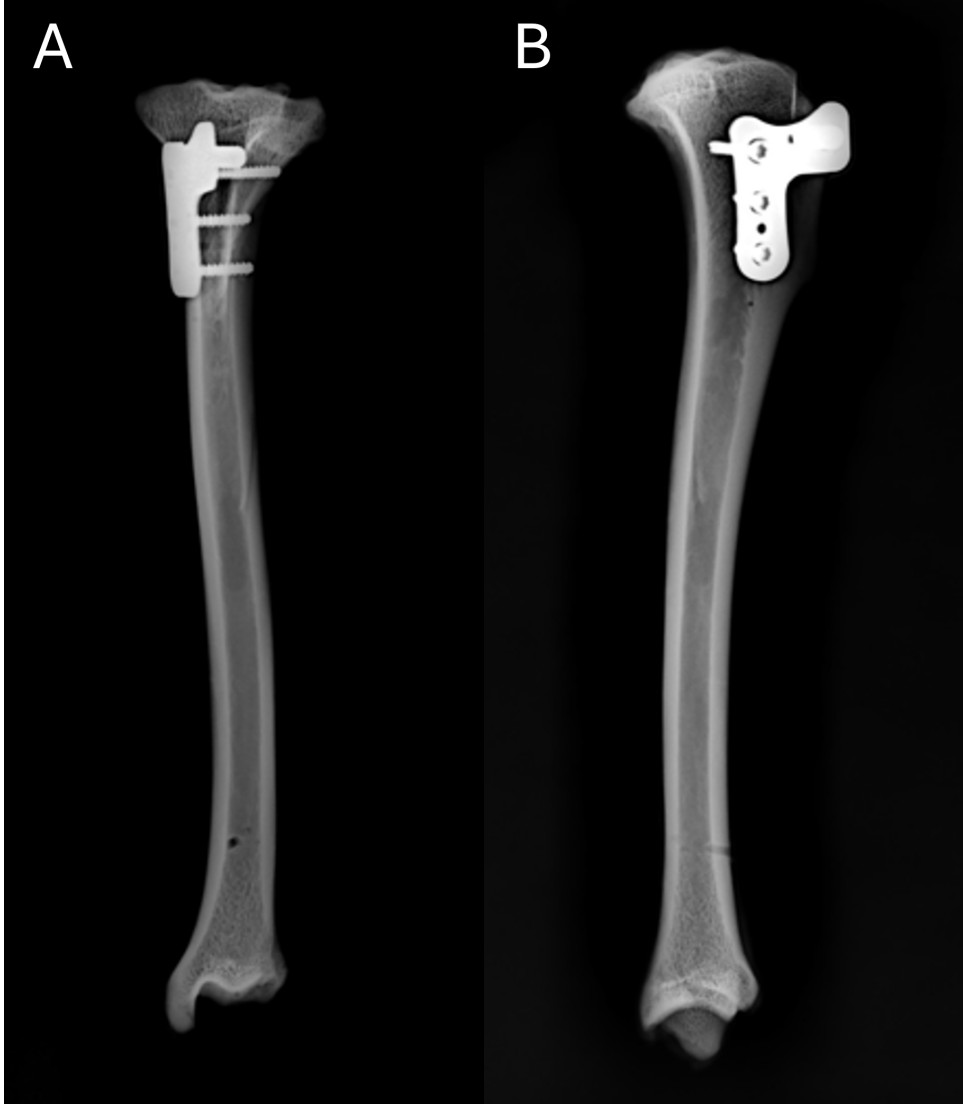

**Fig 6. (A) Radiograph in craniocaudal projection after tibial tuberosity transposition using the RTP implant.** (B) Radiograph in mediolateral projection after TTT using the RTP implant.

was developed to obviate the need for direct fixation to the tibial tuberosity. Its functionality relies on the transposition screw, which prevents the TT from returning to its original position, thereby simplifying the transposition process.

Plates for fixation after performing TTT have demonstrated satisfactory results in correcting tibial tuberosity deviation. However, the tibial shaft must have enough bone stock to support the plate [11]. In our study, all 21 tibias had sufficient bone stock for the placement of three locking screws to secure the plate.

Surgical complications are common in cases of patellar luxation correction using TTT, with studies reporting rates of 10–40% [12,20–22]. The most common complication is patellar reluxation, with rates between 3% and 19.8% [8,14,21]. Other frequent issues include fractures or avulsion of the tibial tuberosity (TTAv), implant failure, and implant migration [8,10,12–14,20]. These are classified as major complications that require surgical revision for correction [8]. In this study,

we did not observe fractures of the tibial crest or implant failure. However, since this was an *ex vivo* study using tibias from canine cadavers, other complications such as patellar reluxation, TTAv, and implant migration were not evaluated.

No bone fissures were detected during the radiographic evaluation conducted in this study. According to Filiquist et al., the manifestation of bone fissures in postoperative radiographs elevates the probability of fracture by 25%, this occurs because the quadriceps exert traction on the tibial tuberosity, subjecting the remaining cortex to cyclic loading, which may ultimately lead to fracture [18].

The literature describes various methods for direct fixation of the transposed TT, with variable complication rates reaching up to 37% in different pin configurations [19,23–25]. A concern regarding TT fixation is the potential for bone fragility induced by the insertion of implants into the tibial crest [17,18,24]. Conversely, the use of an adjacent cortical screw has shown a low complication rate [18]. In this sense, the transposition screw is proposed as an alternative to provide stability without requiring transfixing implants in the TT, as such implants can weaken this structure.

In a study conducted by Rossanese et al. [12], it was reported that distal preservation of the TT is linked to a lower risk of complications. Consistent with this principle, the osteotomy technique used in this study preserves strong bone stock, which is essential for resisting tensile forces [17]. Although our findings are preliminary due to the ex vivo nature of the study, no fractures were observed through visual and tactile inspection or radiographic assessment, and a solid bone bridge was maintained. These initial results are promising and indicate that this approach may help reduce complication rates related to tibial crest fragility in future clinical applications, aligning with the findings of Sullivan et al. [17].

In a study by Sullivan et al. [17], no statistical difference was observed in ultimate failure force and stiffness between three fixation techniques: (i) an incomplete osteotomy fixed with a spacer pin, (ii) two pins and Tension Band Wire (TBW) with complete osteotomy, and (iii) an incomplete osteotomy fixed with two pins into the tibial crest. Although a tensile test was not performed in our study, the transposition screw utilized is designed to prevent segment return, a function analogous to that of a spacer pin, which makes these initial findings promising. Furthermore, when a spacer pin is used, displacement of the tibial crest is performed with a dedicated device, the Tibial Tuberosity Transposition Tool (TTTT); in contrast, the RTP implant utilizes its own transposition screw to achieve this displacement, thereby eliminating the need for an additional device. This study did not include a biomechanical analysis; however, we obtained satisfactory results without any fractures occurring during the transposition process.

The literature shows that implant failure happens before bone healing in 2.1% to 13.8% of cases, leading to unsuccessful TTT [10,13,14,18]. Due to these rates, developing new implants is crucial. *Ex vivo* studies are important initial steps to test implant performance and adaptability, allowing early validation and helping move to biomechanical studies and clinical tests. Therefore, the RTP implant demonstrated good functional performance in this study, confirming its potential for future biomechanical testing.

Among the limitations of this study are primarily the fact that it is a cadaveric study and the variation in bone size, which may impact the results. Although the study confirmed the hypothesis, mechanical and clinical studies are necessary to conclude the device's effectiveness. Another limitation is the absence of a locking mechanism for the transposition screw, which could hypothetically lead to its loosening or back-out before bone healing in live patients. For this reason, the implant will be optimized to prevent such micromotion before bone healing, with a view to future clinical applications. A biomechanical study is required to evaluate the implant, particularly to determine the failure mode of the implant and the tibial crest. In live patients, quadriceps traction can induce implant failure and TTAv, corroborating the need for a clinical study regarding the developed implant. Comparative studies between the use of the developed implant and other tibial tuberosity fixation techniques are also needed to evaluate the advantages and disadvantages associated with the proposed technique.

The implant can be improved; for instance, its plate thickness could be reduced, as it is positioned in an area with minimal muscle coverage, which, hypothetically, could reduce the risk of implant exposure.

It is important to emphasize that this *ex vivo* study demonstrates the implant's capability to perform the transposition in a slow and gradual manner. However, as this study did not include a biomechanical evaluation under loading conditions, it does not describe potential complications that could arise when loads are applied to the implant. Although the current results are promising, this study provides a foundation for future investigations into the implant's efficiency.

## Conclusion

The RTP implant has demonstrated efficacy in facilitating the transposition of the tibial tuberosity in canine cadavers, owing to its capability to gradually and steadily transpose the tuberosity without necessitating supplementary devices.

## Supporting information

**S1 Video. Accelerated video demonstrating tibial tuberosity transposition using the RTP implant.**
(MP4)

## Acknowledgments

The authors would like to thank Daniel Ricardo for helping in the development of the implant.

## Author contributions

**Conceptualization:** Gabriel Rampanelli, Olicies da Cunha.

**Data curation:** Gabriel Rampanelli, Olicies da Cunha, Anderson Luiz de Carvalho.

**Formal analysis:** Gabriel Rampanelli, Olicies da Cunha, Anderson Luiz de Carvalho, Cassio Ricardo Auada Ferrigno.

**Investigation:** Gabriel Rampanelli, Anderson Luiz de Carvalho, Camila Aparecida Luiz, Laura Ayala Lazarotto, Lucas Dill Mocellin, João Pedro Cosmo Machado, Fernando Lunardelli.

**Methodology:** Gabriel Rampanelli, Olicies da Cunha, Anderson Luiz de Carvalho.

**Project administration:** Gabriel Rampanelli, Olicies da Cunha, Anderson Luiz de Carvalho.

**Resources:** Gabriel Rampanelli, Olicies da Cunha, Anderson Luiz de Carvalho, Fernando Lunardelli.

**Supervision:** Olicies da Cunha, Anderson Luiz de Carvalho, Cassio Ricardo Auada Ferrigno.

**Validation:** Olicies da Cunha, Anderson Luiz de Carvalho, Cassio Ricardo Auada Ferrigno.

**Writing – original draft:** Gabriel Rampanelli, Olicies da Cunha, Anderson Luiz de Carvalho, Cassio Ricardo Auada Ferrigno.

**Writing – review & editing:** Gabriel Rampanelli, Olicies da Cunha, Anderson Luiz de Carvalho, Cassio Ricardo Auada Ferrigno.

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
