## [Decision Letter · Decision Letter 0]

5 May 2025

Dear Dr. Ferrigno,

Thank you for submitting your manuscript to PLOS ONE. After careful consideration, we feel that it has merit but does not fully meet PLOS ONE’s publication criteria as it currently stands. Therefore, we invite you to submit a revised version of the manuscript that addresses the points raised during the review process.

A critical point of the manuscript is the following.

The objective describes the desire to evaluate the complications of the proposed TTT technique. However, this static cadaveric study (which does not include a dynamic biomechanical study) has provided as outcomes only the evaluation of the integrity of the tibial tuberosity and the implant immediately after transposition and without any load. Most of the studies cited in discussion and compared with this one (both clinical studies and ex-vivo studies) have evaluated the complications inherent to the integrity of the bone and the implant by providing a load.

Considering that bone integrity after transposition was the primary outcome of the study, a postoperative CT assessment (CT was performed only in pre-surgical planning) could have had a greater sensitivity in detecting small tibial crest fissures.

The described objective of evaluating complications is too generic and inconsistent with what was actually evaluated in the study and may lead the reader to think that this technique does not foresee complications in an absolute sense. It is recommended to be more accurate in describing the objective of the study, the results, the discussion and the conclusions making them consistent with what was actually evaluated.

We look forward to receiving your revised manuscript.

Kind regards,

Adolfo Maria Tambella, DVM, MSc

Academic Editor

PLOS ONE

 [Engevet partially funded the research, which included the manufacturing of the implant and the instruments used].

Additional Editor Comments (if provided):

Reviewers' comments:

Reviewer's Responses to Questions

**Comments to the Author**

1. Is the manuscript technically sound, and do the data support the conclusions?

Reviewer #1: Yes

Reviewer #2: Yes

2. Has the statistical analysis been performed appropriately and rigorously?

Reviewer #1: Yes

Reviewer #2: N/A

3. Have the authors made all data underlying the findings in their manuscript fully available?

Reviewer #1: Yes

Reviewer #2: Yes

4. Is the manuscript presented in an intelligible fashion and written in standard English?

Reviewer #1: Yes

Reviewer #2: No

Reviewer #1: Dear Authors,

Congratulations on your work, which I have read with great interest. The novel implant seems to me to be very interesting to perform tibial trasposition. In addition, these data offer important insights for further studies on mechanical tests and comparison with other surgical techniques. I have few queries/suggestions for changes.

Lines 63-64: The statement does not seem correct to me, I believe that some of the implants described have the ability to dislocate the tibial tuberosity

Line 73-74: The approval of the ethics committee is fine but the criteria for choosing bone explants, the conditions of donors, etc., should be described.

Line 117: I probably didn't understand something well but I have a question: why was it chosen to indicate 2/4 and not the simplification of the fraction (divisione) that is ordinarily used (1/2)?

Lines 119-125: which software was the pre-operative planning performed with?

Line 143: it would be interesting to indicate to readers in terms of the rpm of the screw in order to comply with the speed previously reported in the literature.

Lines 149-150: did you also assess other parameters on the postoperative radiographs or only the presence or absence of fissuring/fracture of the tibial tuberosity?

Lines 160-161: My main concern is about complications. I do not think it is correct to exclude complications due to implant failure or deformation without performing mechanical tests.

Lines 168-169: what should be the indications for implant failure? how can pull-out occur on a fully skeletonised tibia without a load test?

Line 177: “The osteotomy line was shifted caudally”. This statement does not seem to me to be reflected in the attached images, wouldn't it be better to insert an image where this can be seen?

Lines 240-243: how was the weakening of the tibial crest verified?

Lines 257-260: how has implant failure been ruled out?

Figure 6: there are only five captions but a total of six figures.

Reviewer #2: This is generally a well written article, but unfortunately requires some areas of revision to make it more suitable for publication. The text is well supported by pertinent references, and most, if not all, key references appear to have been included. It is an implant that could have an interesting future, but does need further studies (and I hope the authors progress to biomechanical studies).

Language and grammar is generally good, but there are some areas where more appropriate words / phrases should be used, and there are some areas where I suspect language differences and / or autocorrect have created unfortunate errors – easily rectified! Details are given below.

However, there are a few issues that need addressing (more details in body of text):

- the article is perhaps a bit too long – it could (and should) be abbreviated

- there are some areas of superfluous information and a tendency to over-reference

- there are some areas that need further detail (eg on translation screws).

- perhaps most significantly, the authors have made some assumptions / leaps in conclusion that are perhaps a bit optimistic, and make some unfair and/or inaccurate comparisons.

In short, this is a paper that should ultimately be published, but with some new information added, some excess material removed, and the other areas lightly revised and appropriate rewording performed.

Cover:

15-16 were all authors equal contributors? Do we need two lines to state these? Perhaps it is a journal specific requirement - check PONE requirements.

Abstract:

19-21: First sentence – word order is odd (eg tibias in canines cadaers should be after TT) and is it realy investigating potential complications?

24: suggest add in calculated desired transposition Osteotomy was performed

26-28: sentence needs rewording. Specifically the 21 tibias from canine cadavers. Should be earlier in the paragraph to make it read better

Intro:

40-41: – suggest change animals to dogs and delete ref to the stifle.. Add in origin after traumatic.

52: what do you mean by a fluid and dynamic process?

54: if its easy to perform, why the paper?

56: reluxation, not relaxation

59: suggest upto 19.8% or 3-19.8 (which you have quoted elsewhere)

52-60: these paragraphs could probably be combined and abbreviated

62: missing “the”before TT

64: do you mean to inherently displace the TT? Perhaps add ïn situ’after fixed

65: consider ïs desriable”or somehting along those lines rather than considered an option

Study model:

75: how were they selected? Was there any selection for age / size / sex?

79: what are osteotomy errors? Were they incomplete tibias in some cases?

80: was 6deg for 12 hours sufficient for adequate defrosting?

Plate:

84: First sentence superfluous

96: how does the screw engage in the plate and with the bone. Is the screw hole threaded? Was there anything to stop backing out. Were they selected to be just long enough to transpose the bone and then lock in. Were the screws blunt ended or tipped? Need more details. Not necessarily lengthy, but enough to allow the reader to (within reason) replicate

99: what do you mean by implant systems? Do you mean 5 different plate sizes. What about Left vs Right?

114: what CT windowing was employed?

117: one-half (rather than two quarters) would read better

118: at what level was the TT width measured?

119: perhaps better to state what plane was utilised

Implant application:

132: I presume it was a medial approach? Whilst relatively obvious, this is not made clear.

133-134: was it aligned with or just near tubercle of Gerdy? It kind of contradicts what it is in the previous section.

135-136: You do not state an osteotomy was made! How did you measure 80% and 60% while making the cut?

142: “screw was inserted”...where? This sentence could also be made clearer. Eg “..and advanced until the premeasured amount of displacement was achieved”, or something like that

143: how was the rate of advancement measured?

147: how were they assessed – visually? Xray? CT? In particular, how could you assess the integrity without mechanical testing?

Results:

152: how was the size selection made? Eyeball? Measured from the CT? Other?

156: one-half, not 2-quarters

157: probably better to use period rather than comma in decimal figures

162: Table 1 – why were all (bar one) >0.5, when this was your measured target? Not a problem in itself, but may warrant some comment in the text / discussion – eg was it a limitation of measurement, technical difficulty of applying the screw?

Postop evaluation:

167: evaluation of what? Presumably of the operated limbs. What views were taken. Equipment?

Discussion:

177: shifted caudally relative to what?

184: it’s not really deformation per se – deformation is not a force

186: not bone adaptation – this has a specific meaning in bone biology sp preferable to use alternative wording

205-243: this is too long, and contains more detail than really necessary, and so should, in my opinion, be shortened. You also tend to jump around a bit – would be preferable to change the order, grouping comments on say one pin vs 2 pin together. Also, be consistent in pin vs k-wire

224: in rather than on the TC?

241-243: hard to prove there was no weakening You can state there was no fracture / fissure but cannot really state there was no weakening without further testing

244-248: this sentence has no ending – there is no meaning to it. Also, are you mentioning all 3 techniques in the paper? If so, you are missing one (paper is spacer pin vs complete osteotomy + 2 pins and TBW vs partial osteotomy and 2 pins).

253: you say “as a result...” (of the transposition being performed by the implant device) “...there were no complications” but you cannot reasonably conclude that there were no complications due to this difference

257: what do you mean by bone consolidation?

257-260: Its not really a fair comparison. As you rightly acknowledge, there is no biomech analysis. All you can conclude is that no immediate implant failure was noted when the implant was applied

261-266: be cautious in your comparisons here. Whilst plates are involved in both systems (yours and theirs), it is a different method of application, and especially given the lack of detail on your transposition screw and how it engages (or otherwise) with the bone – so cannot be directly compared. It is however worth mentioning the plating systems available (but try not to make it too lengthy).

266-267: unfair comparison – theirs was a clinical caseload vs ex-vivo study. Hard to definitvely compare the lack of fractures in such different models

269: typo: “does not require fixation”

272: delete “for its use”

278-280: we observed...tuberosity. What do you mean to say with this? What is your comparison with xray measurements? Is there a possible reference for this?

282-289: a little long. Good to recognise limitations and potential areas to improve the device and research ideas, but this could be made shorter.

290+: I think this paper merits some further discussion on limitations

293-294: I think I know what you are trying to say, but it reads as though the quads traction during both extension and flexion. Could be improved in wording, And I’m not sure stimulate is the most appropriate word here – perhaps induce?

295: warrants biomech study prior to any clinical study please!

Figures:

319: typo in Ltd

Fig5 B: are you concerned that from a clinical perspective, if a transposition of >50% was required (in this particular imaged bone, or maybe less in others), advancing the transposition screw will actually engage the tibial shaft rather than the TT?

**Do you want your identity to be public for this peer review?** For information about this choice, including consent withdrawal, please see our Privacy Policy

Reviewer #1: No

Reviewer #2: No

---

## [Author Response · Author response to Decision Letter 1]

7 Jul 2025

The objective describes the desire to evaluate the complications of the proposed TTT technique. However, this static cadaveric study (which does not include a dynamic biomechanical study) has provided as outcomes only the evaluation of the integrity of the tibial tuberosity and the implant immediately after transposition and without any load. Most of the studies cited in discussion and compared with this one (both clinical studies and ex-vivo studies) have evaluated the complications inherent to the integrity of the bone and the implant by providing a load.

R: We thank the editor for the correction. In response to the comments and suggestions from the editor and reviewers, the manuscript has been duly revised. We agree with the consideration that the present study was static and was limited to assessing the occurrence of complications at the time of tibial tuberosity displacement. Consequently, the study's objective has been readjusted to strictly reflect the outcomes assessable within its scope. We would like to emphasize, however, that the primary objective behind the development of this implant is to demonstrate its intrinsic capacity to perform tibial tuberosity transposition slowly and gradually. This functionality obviates the need for specific surgical instrumentation or auxiliary devices for this purpose, thereby aiming to simplify the surgical procedure and enhance its precision. To this end, the manuscript has been revised not only to clarify the objective but also to adjust the discussions regarding the assessment of complications (in the Materials and Methods, Results, Discussion and Conclusion sections), particularly concerning the comparison with other biomechanical and clinical studies. Below are some excerpts with the modifications made.

Ln28: The samples underwent radiographic evaluation and manual palpation following the application to detect tibial crest fractures. The plate's shape adequately fit the medial surface of the tibia, and the implant was effective in promoting a slow and gradual transposition in canine cadavers without the need for an additional surgical device to perform the maneuver. The implant proved to be effective in achieving the desired transposition in a progressive, gradual, and slow manner.

Ln64: This study aims to describe the RTP (Rampanelli Transposition Plate) implant developed by the authors in collaboration with the company Engevet (Engevet Ltd, Paraná, Brazil) and to evaluate the TTT technique using the developed implant. We hypothesize that the implant can facilitate tibial tuberosity transposition in a slow, progressive, and gradual manner.

Ln193: The outcomes of this ex vivo investigation substantiated the effectiveness of the RTP implant in executing the transposition maneuver in a slow, progressive, and gradual manner, thereby corroborating the hypothesis.

Ln302: The RTP implant has demonstrated efficacy in facilitating the transposition of the tibial tuberosity in canine cadavers, owing to its capability to gradually and steadily transpose the tuberosity without necessitating supplementary devices.

Q: Considering that bone integrity after transposition was the primary outcome of the study, a postoperative CT assessment (CT was performed only in pre-surgical planning) could have had a greater sensitivity in detecting small tibial crest fissures.

R: Thank you for the comment. We agree that postoperative CT assessment would provide greater sensitivity for detecting small fractures of the tibial crest. However, it was unfortunately not feasible to perform postoperative CT in this study. Therefore, the evaluation was carried out through visual inspection and radiographic imaging. Although radiography has lower sensitivity compared to CT, it remains an effective method for assessing fractures.

Q: The described objective of evaluating complications is too generic and inconsistent with what was actually evaluated in the study and may lead the reader to think that this technique does not foresee complications in an absolute sense. It is recommended to be more accurate in describing the objective of the study, the results, the discussion and the conclusions making them consistent with what was actually evaluated.

R: We thank the Editor for the positive feedback. We agree that, as previously written - particularly after considering the suggestions made by you and the Reviewers - the methodology described in this manuscript could lead readers to assume that no complications are associated with the use of the implant. Aware of the study's limitations, we revised portions of the objectives, results, discussion, and conclusion sections accordingly.

The sections of the manuscript that addressed complications not assessed in this study have been removed. Examples of sections that contained this information and were consequently adjusted are cited below. As this was a common observation from the Editor and Reviewers 1 and 2, all changes made in response to this point are detailed in the respective individual responses to them.

The sections of the manuscript that addressed complications not assessed in this study have been removed. Examples of sections that contained this information and were consequently adjusted are cited below. As this was a common observation from the Editor and Reviewers 1 and 2, all changes made in response to this point are detailed in the respective individual responses to them. Furthermore, to align with the changes made, the study's objective has also been altered.

Ln64: This study aims to describe the RTP (Rampanelli Transposition Plate) implant developed by the authors in collaboration with the company Engevet (Engevet Ltda, Paraná, Brazil) and to evaluate the TTT technique using the developed implant. We hypothesize that the implant can facilitate tibial tuberosity transposition in a slow, progressive, and gradual manner.

R: We have reviewed the manuscript and made all necessary adjustments to ensure it complies with PLOS ONE's style requirements, including proper file naming. We have used the PLOS ONE style templates provided in the link mentioned.

R: Thank you for raising these points. Below are the requested revisions to the Methods section, in accordance with PLOS ONE submission requirements. Additionally, information concerning the ethics committee is detailed on line 71.

(1) Ln 74: The animals died from causes unrelated to this study and were sourced from the Pathology Service of the Federal University of Paraná – Palotina Sector.

[Engevet partially funded the research, which included the manufacturing of the implant and the instruments used].

R: Engevet partially funded the research, exclusively by manufacturing the implant. This study was financed in part by the Coordenação de Aperfeiçoamento de Pessoal de Nível Superior - Brasil (CAPES) – Finance Code 001. Author GR received a research fellowship from CAPES, which supported the development of this study.

b) State what role the funders took in the study.

R: The funders had no role in study design, data collection and analysis, decision to publish, or preparation of the manuscript.

R: None of the authors received any salary or financial compensation from the sponsors of this study.

R: The information regarding the sources of funding has been duly included in the cover letter.

Response to Reviewers

Reviewer #1:

Q: Lines 63-64: The statement does not seem correct to me, I believe that some of the implants described have the ability to dislocate the tibial tuberosity.

R: We thank the reviewer for his/her suggestion. However, we consider that, to perform the transposition maneuver, the implants require the use of an auxiliary device or surgical instrument to displace the tibial tuberosity, such as the Tibial Tuberosity Transposition Tool. Although it is technically possible to use the pin itself as a lever, this approach is not recommended by the authors due to the potential risk of complications, including tibial crest fracture, and the lack of scientific publications reporting or supporting this technique. In the case of implants such as the Rapid Luxation System, a specific tool is available to perform the transposition. In this context, it is understood that the currently available implants do not inherently provide the means for tuberosity displacement, but only for its fixation. We acknowledge that the statement could be perceived as presumptuous; therefore, we have revised its wording to mitigate this impression.

Ln 61: …and require an additional device or surgical instrument to perform the transposition.

Q: Line 73-74: The approval of the ethics committee is fine but the criteria for choosing bone explants, the conditions of donors, etc., should be described.

R: We appreciate your valuable observation regarding the criteria for selecting bone explants and the conditions of the donors. We have included information regarding the age and weight of the animals and clarified that the specimens were obtained from the Pathology Service. After explantation, the bones were evaluated for signs of fractures, neoplasia, and open growth plates, as described “The inclusion criteria were bones without signs of fractures, bone callus, neoplasms, or open growth plates (lines 79-80).

Ln 72: Twenty-one tibias from adult dogs, aged between 1 and 12 years and weighing between 4 and 25 kg of small to medium-sized breeds, regardless of sex, were selected. The animals died from causes unrelated to this study and were sourced from the Pathology Service of the Federal University of Paraná – Palotina Sector.

Q: Line 117: I probably didn't understand something well but I have a question: why was it chosen to indicate 2/4 and not the simplification of the fraction (divisione) that is ordinarily used (1/2)?

R: Thank you for pointing out this detail. In other referenced studies, the value of two-quarters was used. However, we have decided to follow your suggestion and change it to one-half, as it is the standard and more commonly used notation. We have also made the corresponding adjustment in the Results section to ensure consistency with the proposed change.

Ln 122: …the transposition distance was standardized at one-half of the width of the tibial tuberosity, measured at the level of the patellar tendon insertion in the frontal plane.

Ln 173: Transposition required to achieve an equivalent to one-half of the width of the tibial tuberosity varied…

Q: Lines 119-125: which software was the pre-operative planning performed with?

R: Thank you for this important observation. We have added the information that the surgical planning was performed using the vPOP Pro software.

Ln 124: …within the vPOP Pro (VETSOS Education Ltd. – Shrewsbury, United Kingdom) software …

Q: Line 143: it would be interesting to indicate to readers in terms of the rpm of the screw in order to comply with the speed previously reported in the literature.

R: We thank the reviewer for the observation, have rewritten the sentence, and added information regarding the RPM. We consider that the revolutions per minute refer to the pitch of the transposition screw, and therefore, we have omitted this information from the text.

Ln 157: This process was executed slowly and gradually, at a rate of 0.7 mm/min, accompanied by a rotational speed of 1.4 revolutions per minute (RPM) for the M3x0.5 transposition screw and 1 RPM for the M4x0.7 screw [17,18].

Q: Lines 149-150: did you also assess other parameters on the postoperative radiographs or only the presence or absence of fissuring/fracture of the tibial tuberosity?

R: Thank you for the question. We also evaluated the implant placement and screw positioning radiographically. We rewrote the sentence structure to make it clearer, as the previous version did not properly convey that these assessments were performed both visually and radiographically. Additionally, another reviewer suggested including the specifications of the equipment used for performing the radiographic examinations.

Ln163: Upon completion of the implant application, the samples underwent visual and tactile inspection, as well as radiographic evaluation (Diafix Ltd, 500mA/125 kV - São Paulo, Brazil) in both mediolateral and craniocaudal projections. This was conducted to verify the displacement of the tibial tuberosity, identify any potential fractures of the tibial tuberosity, and assess the positioning of the implant and screws.

Q: Lines 160-161: My main concern is about complications. I do not think it is correct to exclude complications due to implant failure or deformation without performing mechanical tests.

R: We thank the reviewer for the comment. We share the same concern regarding potential complications. As mentioned, it is not possible to rule out implant failure or structural deformation without conducting specific mechanical testing. Considering this, we revised the presentation of the results to ensure that the conclusions are limited to what can be reliably stated based on the in situ assessment of the samples. We also revised the objectives, results, discussion, and conclusion sections to ensure consistency and alignment with the modifications made throughout the manuscript.

Ln175: All tibias from cadavers reached the stipulated transposition in a slow, gradual, and controlled manner. Furthermore, appropriate positioning of the transposition screw was observed near the insertion of the patellar ligament.

Ln179: In all specimens, the plate showed proper placement on the proximal tibial shaft, with sufficient bone stock to accommodate three fixation screws. Moreover, all screws remained well positioned. All specimens underwent visual and tactile inspection after the intended displacement, and no mobility suggestive of fracture was detected.

Q: Lines 168-169: what should be the indications for implant failure? how can pull-out occur on a fully skeletonised tibia without a load test?

R: We thank the reviewer for the suggestion and correction. We apologize for the mistake. We agree with your statement — it was a well-noted observation. Without mechanical load testing, it is not possible to detect implant failure or screw pull-out, as no force is applied to the system. Considering these aspects, we revised the sentence to clearly state what could actually be evaluated.

Ln186: Additionally, proper positioning of the screws and plate was observed, as well as the displacement of the tibial tuberosity.

Q: Line 177: “The osteotomy line was shifted caudally”. This statement does not seem to me to be reflected in the attached images, wouldn't it be better to insert an image where this can be seen?

R: We thank the reviewer for the correction. We were referring to the amount of remaining bone stock of the tibial tuberosity, which, when compared to early reports (ex: DeAngelis, 1971), shows a greater amount of available bone stock. We revised the sentence to more clearly convey the intended message to the reader. Furthermore, we have made an addition to the text (at the end of the paragraph in question) to enhance the clarity of the pertinent information.

Ln199: The incom

---

## [Decision Letter · Decision Letter 1]

1 Sep 2025

Dear Dr. Ferrigno,

Thank you for submitting your manuscript to PLOS ONE. After careful consideration, we feel that it has merit but does not fully meet PLOS ONE’s publication criteria as it currently stands. Therefore, we invite you to submit a revised version of the manuscript that addresses the points raised during the review process.

We look forward to receiving your revised manuscript.

Kind regards,

Adolfo Maria Tambella, DVM, MSc

Academic Editor

PLOS ONE

Journal Requirements:

Reviewers' comments:

Reviewer's Responses to Questions

**Comments to the Author**

Reviewer #1: All comments have been addressed

Reviewer #2: (No Response)

2. Is the manuscript technically sound, and do the data support the conclusions?

Reviewer #1: Yes

Reviewer #2: Yes

3. Has the statistical analysis been performed appropriately and rigorously?

Reviewer #1: N/A

Reviewer #2: N/A

4. Have the authors made all data underlying the findings in their manuscript fully available?

Reviewer #1: Yes

Reviewer #2: Yes

5. Is the manuscript presented in an intelligible fashion and written in standard English?

Reviewer #1: Yes

Reviewer #2: Yes

Reviewer #1: I believe that the answers are satisfactory, only one doubt remains. The study is static as it is performed on anatomical pieces, but when the implant is mounted on patients who will use the limb, what element will be able to guarantee that the transposition screw maintains the level of depth chosen at the time of surgery and does not tend to change its position (unscrew), before healing, with motor activity?

If there are no blocking elements, this aspect should be hypothesized and indicated among the limits of this study

Reviewer #2: Thank you for your revision and addressing our comments and suggestions.

While the paper has been improved significantly, there are still some areas that require some (minor) revisions. Suggested changes are listed below. Most are just suggested minor word changes to help readability, but there are some areas of possible repetition that could be combined and condensed.

L20. reads oddly - suggest delete the "in 21 cadaveric canine tibias". Could be included instead in L23

L24. calculate

L24 suggest add in osteotomy "of the tibial tuberosity"

L41 can be of either...

L61 but what about digital / manual transposition

L62 don't need " "around in situ; perhaps 'applied' rather than fixed?

L85 suggest comprising rather than formed by

L107 suggest delete meticulously

L130 suggest delete strategically

L143 suggest delete "After planning...began" (superfluous)

L153 Labels missing from images supplied

L179-181 Seems to be repeating content, unless you mean specifically about assessment post procedure - if so, would benefit from clearer wording

L198 mitigates not really appropriate here.

L199-200 repetition of content re crest intact - could be condensed with earlier line on this

L202 do you mean applied rather than resultant force?

L227 suggest reword - the fissure doesn't permit the force - he force is being applied regardless.

L237-242 Again, repeats about the bone stock . width - suggest combine with earlier part of discussion

L242-244. unfortunately this does not prove it counteracts tensile forces, as no mechanical studies have been performed

L268-279 Feels out of place. Perhaps move earlier in discussion, before discussing implant failure etc

L284 reword. "about a tensile test" doesn't really read correctly. Perhaps delete "about...designed"

L287 avTT - different acronym used elsewhere for this (L218)

L297 repetition of the biomech evaluation

Fig 4 - need letters added in (L153)

**Do you want your identity to be public for this peer review?** For information about this choice, including consent withdrawal, please see our Privacy Policy

Reviewer #1: No

Reviewer #2: No

---

## [Author Response · Author response to Decision Letter 2]

1 Oct 2025

Dear Academic Editor and Reviewers

RE: PONE-D-25-12917 “Evaluation of a New Implant For Tibial Tuberosity Transposition in Dogs: An Ex Vivo Study”

We would like to thank for the opportunity to submit a revised version of our manuscript, and we believe that the referees presented thoughtful and constructive comments on our manuscript, and we are comfortable in addressing each one in this revised version.

Please find below responses to each comment. Changes in the manuscript are highlighted in red in a ‘tracked’ file. The highlighted version and a clean version of the revised manuscript were uploaded to the PLOS One submission site. All authors have read and approved the revised version of the manuscript as submitted.

During the process of revising the manuscript in response to the suggestions from the Editor and the Reviewers, we identified additional points where the text could be improved for greater clarity and readability. We have taken the initiative to make these supplementary corrections, which are duly highlighted in the text. However, we would like to emphasize that should the Academic Editor and the Reviewers prefer the previous version, we are entirely willing to revert these modifications.

We hope that you find our responses satisfactory. Thank you for your consideration of our manuscript. If you need any additional information, please, do not hesitate to contact us.

Yours sincerely,

The Authors

Evaluation of a New Implant for Tibial Tuberosity Transposition in Dogs: An Ex Vivo Study

Response to Reviewers

Comments: During the process of revising the manuscript in response to the suggestions from the Editor and the Reviewers, we identified additional points where the text could be improved for greater clarity and readability. We have taken the initiative to make these supplementary corrections, which are duly highlighted in the text. However, we would like to emphasize that should the Academic Editor and the Reviewers prefer the previous version, we are entirely willing to revert these modifications.

Ln51: The TTT is an effective technique for realigning the quadriceps extensor mechanism and can be combined with other procedures [9, 10].

Ln87: A custom inverted 'L'-shaped locking plate was created using Autodesk Fusion 360 (Autodesk Inc., San Francisco, USA).

Ln90: The design allows for secure attachment of the implant to the tibial shaft, with one designated hole strategically positioned on the tibial tuberosity to improve the transposition process.

Ln101: Considering the wide range of sizes among dogs, five implant sizes were recommended, with distinct variations for the right and left sides. After the design process, both the plate and fixation screw were made from titanium (ASTM F163), while the transposition screw was made from 316L stainless steel.

Ln123: The tibial mechanical axis was initially delineated in the medial view of the surgical plan within the vPOP Pro (VETSOS Education Ltd. – Shrewsbury, United Kingdom) software.

Ln132: The tibial crest measurement determined values corresponding to 80% of the osteotomy line on the medial surface and 60% on the lateral surface.

Ln162: Upon completion of the implant application, the samples underwent visual and tactile inspection and radiographic evaluation…

Ln219: Surgical complications are common in cases of patellar luxation correction using TTT, with studies reporting rates of 10 to 40% [12,20-22]. The most common complication is patellar reluxation, with rates between 3% and 19.8% [8,14,21]. Other frequent issues include fractures or avulsion of the tibial tuberosity (TTAv), implant failure, and implant migration [8,10,12-14,20]. These are classified as major complications that require surgical revision for correction [8]. In this study, we did not observe fractures of the tibial crest or implant failure. However, since this was an ex vivo study using tibias from canine cadavers, other complications such as patellar reluxation, TTAv, and implant migration were not evaluated.

Ln246: These initial results are promising and indicate that this approach may help reduce complication rates related to tibial crest fragility in future clinical applications, aligning with the findings of Sullivan et al. [17].

Ln262: The literature shows that implant failure happens before bone healing in 2.1% to 13.8% of cases, leading to unsuccessful TTT [10,13,14,18]. Due to these rates, developing new implants is crucial. Ex vivo studies are important initial steps to test implant performance and adaptability, allowing early validation and helping move to biomechanical studies and clinical tests. Therefore, the RTP implant demonstrated good functional performance in this study, confirming its potential for future biomechanical testing.

Reviewer #1:

I believe that the answers are satisfactory, only one doubt remains. The study is static as it is performed on anatomical pieces, but when the implant is mounted on patients who will use the limb, what element will be able to guarantee that the transposition screw maintains the level of depth chosen at the time of surgery and does not tend to change its position (unscrew), before healing, with motor activity?

If there are no blocking elements, this aspect should be hypothesized and indicated among the limits of this study

R: We thank the reviewer for the positive feedback and the pertinent question. Indeed, during the conduction of the study, we also considered the possibility of screw loosening. To mitigate this potential complication, we are collaborating with the responsible design engineer on an optimization of the implant design. The main modification will consist of optimizing the thread profile geometry, aiming to increase friction and maximize resistance to loosening under axial load. We acknowledge the validity of this concern and, therefore, the possibility of screw loosening has been included as one of the limitations in the discussion of our study.

Ln272: Another limitation is the absence of a locking mechanism for the transposition screw, which could hypothetically lead to its loosening or back-out before bone healing in live patients. For this reason, the implant will be optimized to prevent such micromotion before bone healing, with a view to future clinical applications.

Reviewer #2:

Thank you for your revision and addressing our comments and suggestions.

While the paper has been improved significantly, there are still some areas that require some (minor) revisions. Suggested changes are listed below. Most are just suggested minor word changes to help readability, but there are some areas of possible repetition that could be combined and condensed.

R: We thank the reviewer for their positive and encouraging feedback. In accordance with your guidance, we have implemented all indicated corrections to improve the manuscript's readability. Sections with potential repetitions have likewise been adjusted and condensed. We greatly appreciate your acknowledgment that the text has significantly improved, and we hope that this revised version meets an even higher standard of quality. The specific changes are detailed in the point-by-point responses to each of your comments.

L20. reads oddly - suggest delete the "in 21 cadaveric canine tibias". Could be included instead in L23.

R: We thank you for the correction. The information in question has been duly added to the indicated sentence.

Ln22: Computed tomography was performed on 21 cadaveric canine tibias to plan the surgical technique and calculate the desired transposition.

L24. Calculate.

R: We thank you for the correction. The term in question has been duly corrected.

Ln23: … calculate…

L24 suggest add in osteotomy "of the tibial tuberosity".

R: We thank the reviewer for the suggestion. The term in question has been added to the corresponding sentence.

Ln24: …an osteotomy of the tibial tuberosity was performed…

L41 can be of either...

R: We thank you for the correction. The term in question has been duly corrected.

Ln40: …can be of either…

L61 but what about digital / manual transposition.

R: We thank the reviewer for the valuable question. Indeed, manual transposition of the tibial tuberosity is a viable approach, harkening back to the first TTT methods described in the literature. However, in recent years, a growing trend towards the use of specific devices to ensure greater procedural precision has been observed. The manual approach can result in lower precision due to micromovement inherent to the manipulation. For this reason, we have chosen to focus on describing implants and methods that represent ‘innovations’ in the technique, but we have adopted the recommendation, and the manuscript has been revised to include information on the manual transposition method.

L61: …additional device or manual pressure to perform the transposition.

L62 don't need " "around in situ; perhaps 'applied' rather than fixed?

R: We thank the reviewer for the suggestion. The corresponding sentence has been duly altered in the manuscript.

Ln62: …implant applied after…

L85 suggest comprising rather than formed by.

R: We thank the reviewer for the suggestion. The corresponding sentence has been duly altered in the manuscript.

Ln85: …comprising a plate for fixation…

L107 suggest delete meticulously.

R: We thank the reviewer for the suggestion. The corresponding sentence has been duly altered in the manuscript.

Ln106: ...was engineered utilizing…

L130 suggest delete strategically.

R: We thank the reviewer for the suggestion. The corresponding sentence has been duly altered in the manuscript.

Ln129: …was drawn…

L143 suggest delete "After planning...began" (superfluous).

R: We thank you for the suggestion. The sentence in question has been removed from the manuscript. As the sentence was deleted in its entirety, it is not possible to demonstrate this change visually, however, that the removal was made on Ln142

L153 Labels missing from images supplied.

R: We thank the reviewer for the correction. The missing caption for Figure 4 has been added, and the sentence in question has been corrected. Additionally, to improve clarity and visual correspondence, the labels in the figure have been changed from letters to symbols.

Ln152: To achieve this, the proximal locking screw was initially inserted, succeeded by the distal locking screw, and lastly, the central locking screw (Fig. 4B).

Ln318: Figure 4. (A) Positioning of the osteotomy guide with two Kirschner wires. (B) Fixation of the RTP implant at the height of the patellar tendon insertion with three locking screws, inserted in sequential order: proximal (arrowhead), distal (asterisk), and central (arrow).

L179-181 Seems to be repeating content, unless you mean specifically about assessment post procedure - if so, would benefit from clearer wording

R: We thank the reviewer for the guidance. Indeed, we acknowledge the presence of repetitive information in the text. Therefore, the sentence in question has been removed to eliminate the redundancy, as can be seen on Ln177.

L198 mitigates not really appropriate here.

R: We thank the reviewer for the correction. The term has been duly replaced with a more suitable alternative in the manuscript.

Ln193: …while the implant maintains the lateral deviation…

L199-200 repetition of content re crest intact - could be condensed with earlier line on this.

R: We thank the reviewer for the suggestion. In response, the sentence in question has been removed, and the paragraph has been restructured to eliminate repetition.

Ln192: The technique employed an incomplete osteotomy, aimed at preserving a bony bridge in the distal [17], while the implant maintains the medial deviation of the newly transposed bone fragment.

L202 do you mean applied rather than resultant force?

R: We thank the reviewer for this question. The reviewer is correct; we were indeed referring to the "applied force." The sentence has been revised.

Ln196: …a greater applied force…

L227 suggest reword - the fissure doesn't permit the force - he force is being applied regardless.

R: We thank the reviewer for the correction. The sentence in question has been rephrased.

Ln230: …this occurs because the quadriceps exert traction on the tibial tuberosity, subjecting the remaining cortex to cyclic loading, which may ultimately lead to fracture [18].

L237-242 Again, repeats about the bone stock . width - suggest combine with earlier part of discussion.

R: We thank the reviewer for the positive feedback. We would like to clarify the logical structure of these paragraphs in the 'Discussion' section. The paragraph in question focuses specifically on the potential of incomplete osteotomy to reduce complications, based on the findings of Rossanese et al. (2019) and Sullivan et al. (2024). In contrast, the preceding paragraph, although also mentioning incomplete osteotomy, discusses the transposition technique itself and the implant's capability to perform it.

We agree that there was some repetitiveness in the original wording. To address this, we have readjusted the sentences to eliminate redundancies and improve the transition, as also suggested in another point (Ln192). However, we respectfully disagree with the suggestion to combine the two paragraphs. We believe that merging them would impair the clarity of the manuscript by conflating two distinct topics (the technique and its complications), which would leave the argumentation about complications displaced from its logical context. Therefore, we have opted to keep the paragraphs separate but with improved wording to avoid repetition

Ln192: The technique employed an incomplete osteotomy, aimed at preserving a bony bridge in the distal region [17], while the implant maintains the lateral deviation of the newly transposed bone fragment.

Ln240: In a study conducted by Rossanese et al. [12], it was reported that distal preservation of the TT is linked to a lower risk of complications. Consistent with this principle, the osteotomy technique used in this study preserves strong bone stock, which is essential for resisting tensile forces [17]. Although our findings are preliminary due to the ex vivo nature of the study, no fractures were observed through visual and tactile inspection or radiographic assessment, and a solid bone bridge was maintained.

L242-244. unfortunately this does not prove it counteracts tensile forces, as no mechanical studies have been performed.

R: We thank the reviewer for the correction. Indeed, we acknowledge that the statement cannot be substantiated within the scope of this work, as no mechanical tests were performed. Therefore, the sentence in question has been readjusted.

Ln245: and a solid bone bridge was maintained.

L268-279 Feels out of place. Perhaps move earlier in discussion, before discussing implant failure etc.

R: We thank the reviewer for the suggestion. In response, the paragraphs in question have been repositioned in the text to precede the discussion of complications. As a result of repositioning the paragraphs in question, the order of the references in the text has been altered. Throughout the manuscript, the references affected by this change have been highlighted in red on their respective lines (L208, L220, L223, L235, L237).

Ln207: Literature reports describe the stabilization of TTT using plates, notably involving fixation of the plate into the tibial crest [11,19]. One example is the Rapid Luxation System (RLS) implant (Rita Leibinger Medical, Tuttlingen, Germany) [11], which employs a spacer secured with screws to the tibial crest to maintain the transposition. In contrast, the RTP implant was developed to obviate the need for direct fixation to the tibial tuberosity. Its functionality relies on the transposition screw, which prevents the TT from returning to its original position, thereby simplifying the transposition process.

Ln215: Plates for fixation after performing TTT have demonstrated satisfactory results in correcting tibial tuberosity deviation. However, the tibial shaft must have enough bone stock to support the plate [11]. In our study, all 21 tibias had sufficient bone stock for the placement of three locking screws to secure the plate.

Ln399:

1. Eskelinen EV, Suhonen AP, Virolain

---

## [Editor Report · Decision Letter 2]

8 Oct 2025

Evaluation of a New Implant for Tibial Tuberosity Transposition in Dogs: An Ex Vivo Study

PONE-D-25-12917R2

Dear Dr. Ferrigno,

We’re pleased to inform you that your manuscript has been judged scientifically suitable for publication and will be formally accepted for publication once it meets all outstanding technical requirements.

Kind regards,

Adolfo Maria Tambella, DVM, MSc

Academic Editor

PLOS ONE

Additional Editor Comments (optional):

The manuscript has been entirely and carefully revised. The current version can be considered suitable for publication in PLoS ONE. Congratulations to the authors! 
---

## [Editor Report · Acceptance letter]

PONE-D-25-12917R2

PLOS ONE

Dear Dr. Ferrigno,

I'm pleased to inform you that your manuscript has been deemed suitable for publication in PLOS ONE. Congratulations! Your manuscript is now being handed over to our production team.

Kind regards,

on behalf of

Prof. Adolfo Maria Tambella

Academic Editor

PLOS ONE